# DETECTING OUT-OF-CONTEXT MISINFORMATION VIA MULTI-AGENT AND MULTI-GRAINED RETRIEVAL

## ABSTRACT

Misinformation remains a critical issue in today's information landscape, significantly impacting public perception and behavior. Among its various forms, out-of-context (OOC) misinformation is particularly pervasive, misrepresenting information by repurposing authentic images with false text. Traditional OOC detection methods often rely on coarse-grained similarity measures between image-text pairs, which fall short of providing interpretability and nuanced understanding. Conversely, whereas multimodal large language models (MLLMs) exhibit vast knowledge and an inherent ability for visual reasoning and explanation generation, they remain deficient in the complexity required to understand and discern nuanced cross-modal distinctions thoroughly. To address these challenges, we propose MACAW, a retrieval-based approach that indexes external knowledge, focusing on multiple granularities by extracting and cataloging relevant events and entities. Our framework first extracts multi-granularity information to assess the contextual integrity of news items, followed by a multi-agent reasoning process for accurate detection. Extensive experiments demonstrate the robustness and effectiveness of our proposed framework in identifying out-of-context fake news, outperforming the state-of-the-art solutions by **4.3%**.

## 1 INTRODUCTION

The exponential growth of social media platforms has dramatically increased the accessibility, cost-efficiency, and speed of news dissemination through multimodal channels (Akhtar et al., 2023). However, this has also led to an increase in the spread of misleading or fabricated information. One particularly effective method of misinformation is **out-of-context (OOC)** news (Qi et al., 2024; Papadopoulos et al., 2024), which manipulates authentic images by presenting them in incorrect or misleading contexts. For example, during the recent U.S. presidential election, malicious actors could have misled voters by swapping current authentic election images with unrelated textual context, thereby creating a false narrative. To address these challenges, two methods have been widely adopted: traditional similarity-based methods and more recent efforts utilizing MLLMs.

**OOC Detection with News Content (Image-Text) Comparison.** Traditional research (Zhou et al., 2020; Abdelnabi et al., 2022; Jaiswal et al., 2017; 2019) on OOC primarily relies on computing similarity scores between image-text pairs or learning unified latent representation spaces. As shown in Figure 1(a), these methods may yield high similarity scores, falsely suggesting a match between the image and text, while failing to detect nuanced discrepancies. The absence of interpretability further limits their utility in tasks requiring detailed verification or reasoning.

**OOC Detection with MLLMs.** Recent efforts (Mu et al., 2023; Qi et al., 2024; Hu et al., 2024; Liu et al.; Wang et al., 2024) have shifted towards using Multimodal Large Language Models (MLLMs) for OOC misinformation detection, aiming to generate predictions with explanations. Despite advancements, even state-of-the-art (SOTA) models like GPT-4o exhibit significant limitations when facing *ambiguous* or *up-to-date* information. As shown in Figure 1(b), GPT-4o, when provided with insufficient context in the image, cannot confirm whether an out-of-context mismatch exists. This reveals a fundamental issue: constrained by the explicit data they receive (Hu et al., 2022), MLLMs lack the deep reasoning necessary for cross-referencing temporal, contextual, or entity details, leading them to default to non-committal conclusions when faced with subtle inconsistencies. Consequently,

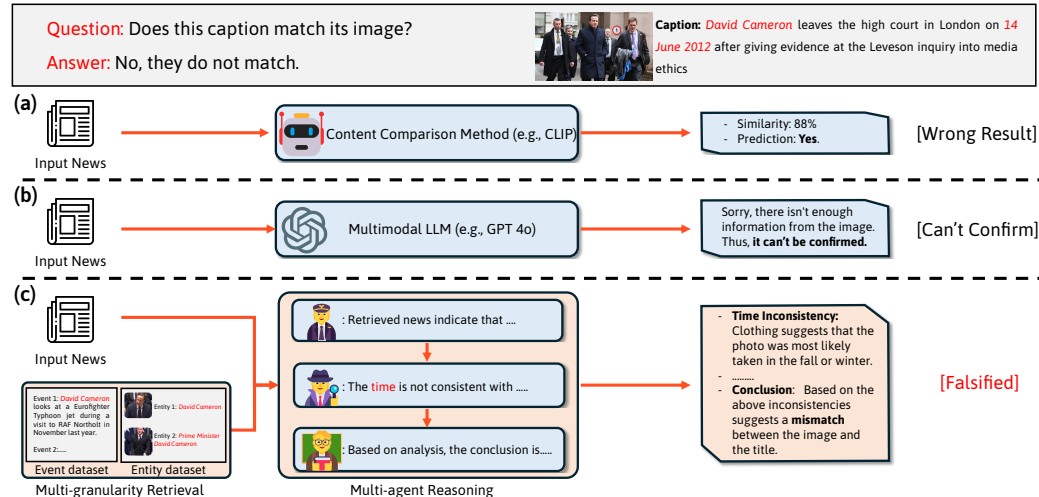

Figure 1: Comparison of existing methods ((a) and (b)) with our multi-agent OOC detection framework (c) in OOC detection task. In light of space constraints, we slightly modified the models' responses for conciseness, ensuring their original meaning remained intact.

while MLLMs-based solutions offer more interpretability than traditional methods, they still fail to detect the intricate discrepancies required for accurate OOC detection.

**Rethinking OOC Detection: How Human Specialists Handle It?**    As mentioned before, existing methods fail to capture the full complexity of OOC detection, particularly the need for **deep, multi-granular reasoning and interpretable, fine-grained explanations**. In practice, fact-checking experts follow a systematic verification process: they retrieve information from multiple sources and modalities, validate details, and reason about timelines, contexts, and inconsistencies (Holan, 2018; Center, 2020). While multiple experts may independently analyze content before reaching consensus, each expert performs this complete sequence of analytical steps.

**Our Proposal: A Multi-Agent Approach with Multi-Grained Retrieval.**    Drawing inspiration from the systematic verification steps that experts perform, we propose **MACAW** (**M**ulti-**A**gent **C**ross-Modal Misinformation **A**nalysis **W**orkflow), a novel multi-agent system for OOC detection and explanation. As shown in Figure 1(c), our system is supported by a self-constructed, multi-granularity database, which integrates information from various levels to offer a robust foundation for detecting and explaining OOC misinformation. MACAW decomposes the sophisticated verification process into optimized subtasks, implementing three specialized agents: the **Retrieval Agent** for gathering relevant information, the **Detective Agent** for examining potential inconsistencies, and the **Analyst Agent** for synthesizing findings and generating final conclusions. Through this structured division of responsibilities, our architecture ensures comprehensive verification while maintaining both efficiency and interpretability in identifying subtle contextual discrepancies.

**Contributions.**    Our contribution can be summarized as follows:

- We construct a **self-constructed multi-granularity database** for OOC detection, which encapsulates both entity and event-level information of existing news and knowledge.

- We propose a **multi-agent** OOC detection framework MACAW, which cross-validates multi-granularity information with input news. It can not only perform sophisticated reasoning and OOC detection, but also give **interpretation** that the news is OOC based on which information source.

- Extensive experiments validate the robustness and effectiveness of our framework across various types of OOC misinformation. Our framework achieves a **4.3%** improvement in accuracy compared to the SOTA methods, demonstrating its superior performance in detecting OOC misinformation.

## 2 RELATED WORK

### 2.1 KNOWLEDGE-ENHANCED MISINFORMATION DETECTION

Early misinformation detection research focused on semantic feature extraction from news content, but as fake and real news became semantically indistinguishable, researchers shifted towards leveraging external knowledge (Zhou & Zafarani, 2020). This transition led to various knowledge-enhanced approaches, such as CompareNet (Hu et al., 2021), which constructs directed heterogeneous document graphs to compare news content with knowledge bases through entity extraction. Building on this foundation, recent work has emphasized knowledge retrieval for more precise fact-checking. Notable advances include a retrieval-augmented generation framework for evidence-grounded outputs (Yue et al., 2024), a unified inference framework integrating multiple evidence sources (Wu et al., 2024), and document-level claim extraction methods (Deng et al., 2024). While these approaches have demonstrated the value of external knowledge in improving detection accuracy (Dun et al., 2021; Hu et al., 2021; Qian et al., 2021), they have yet to fully address the utilization and interaction between information at different granularities.

### 2.2 VISION-LANGUAGE-MODELS ASSISTED MISINFORMATION DETECTION

While traditional approaches to misinformation detection have predominantly focused on unimodal data, recent advances in vision-language models have significantly enhanced the ability to detect multimodal inconsistencies. For instance, Abdelnabi et al. (2022) extended the NewsCLIPpings dataset (Luo et al., 2021) by incorporating external evidence and introduced the Consistency Checking Network (CCN), which evaluates both image-to-image and text-to-text consistency. Similarly, the Stance Extraction Network (SEN) (Yuan et al., 2023) builds on the same encoders but improves performance by clustering external evidence semantically to infer its stance towards the claim. SEN also enhances consistency detection by capturing the co-occurrence of named entities across textual and external evidence.

Further advancing the field, the Explainable and Context-Enhanced Network (ECENet) combines coarse- and fine-grained attention mechanisms to model multimodal feature interactions (Zhang et al., 2024). ECENet utilizes different encoders to jointly process textual and visual entities, offering more nuanced detection of inconsistencies. In addition, SNIFFER (Qi et al., 2024) addresses both "internal consistency" in image-text pairs and "external consistency" with external evidence. A parallel line of research has focused on developing interpretable multimodal architectures for misinformation detection. These approaches emphasize transparent decision-making processes while maintaining high detection accuracy. Notable works include Liu et al. (2023b); Ma et al. (2024); Zhang et al. (2023b)

## 3 METHODOLOGY

We propose MACAW (**M**ulti-Agent **C**ross-Modal **M**isinformation **A**nalysis **W**orkflow), a multi-grained framework for OOC detection, integrating both fine-grained entity-level and coarse-grained event-level information. As shown in Figure 2, our approach consists of three core components: **1) Evidence Storage**, where visual and textual entities are extracted and aligned using a lightweight MLLM, alongside the storage of event-level information extracted from news captions; **2) Evidence Retrieval**, which retrieves both entity-level and event-level data through a unified encoding mechanism; and **3) Multi-Agent Detection**, leveraging specialized agents to analyze the consistency of the retrieved evidence and generate explainable OOC detection results.

### 3.1 EVIDENCE STORAGE

The evidence storage module is designed to extract, align, and store visual and textual entities, as well as event-level information from news items for efficient retrieval using similarity search of Faiss (Douze et al., 2024). Both visual and textual inputs are processed through specialized models, and only aligned entities are stored for rapid querying.

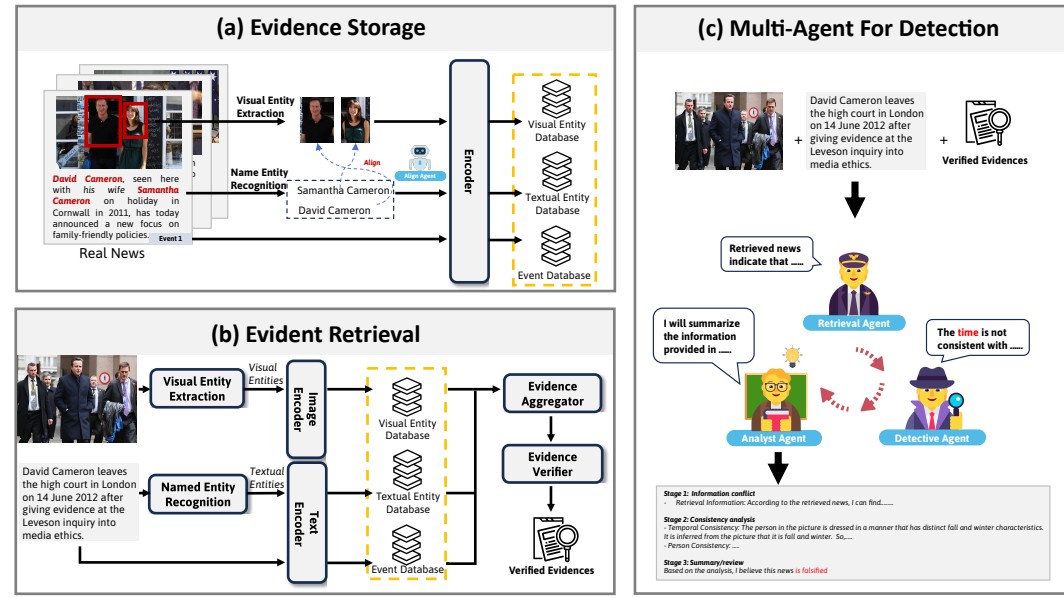

Figure 2: Architecture of the proposed framework MACAW.

### 3.1.1 MULTIMODAL ENTITY EXTRACTION

Given a news item $N = (I, T)$, where $I$ represents the news image and $T$ is the news caption, the system first extracts visual and textual entities. A multimodal entity is defined as a pair consisting of a visual entity and its corresponding textual entity, where both refer to the same real-world object or concept. Specifically, a multimodal entity is represented as $(v_i, t_i)$, where $v_i$ is a visual entity extracted from $I$, and $t_i$ is a textual entity extracted from $T$. Visual entity $v$ is extracted from $I$ using the YOLO v8 Instance segmentation model $M_{\text{YOLO}}$ (Jocher et al., 2023), producing a set of detected visual entities $V$. Textual entity $t$ is extracted from $T$ using the spaCy NER model $M_{\text{NER}}$ (Honnibal & Montani, 2017), resulting in a set of textual entities $T$. Thus, the sets of textual entities and textual entities make up the entity set $E = \{(v_1, t_1), ...., (v_k, t_k)\}$, where $k$ presents the number of entities.

### 3.1.2 MULTI-MODAL ALIGNMENT

Before encoding, the system performs multimodal alignment using a lightweight MLLM. Considering factors such as computational cost, accuracy, and cross-modal understanding capabilities, we selected GPT-4o mini as the model. This model strikes a balance between efficiency and performance, offering robust cross-modal alignment while maintaining low cost. The alignment model $M_{\text{align}}$ gives the similarity between extracted visual entity $v_i$ and textual entity $t_i$, establishing potential mappings between them:

$$S(v_i, t_i) = M_{\text{align}}(v_i, t_i)$$

A mapping between a visual entity $v_i$ and a textual entity $t_i$ is considered valid if the similarity score $S(v_i, t_i)$ exceeds a predefined threshold $\tau$. Only entities with valid mappings are retained for further encoding and storage. Entities without sufficient cross-modal similarity are discarded:

$$E_i = (v_i, t_i) \in E \quad \text{if} \quad S(v_i, t_i) \geq \tau$$

This alignment ensures that only meaningful and relevant visual-textual entity pairs are processed further, reducing storage overhead and improving retrieval precision. The mapping information, along with the aligned entities, is saved for future retrieval and analysis.

### 3.1.3 Encoding and Storage

After establishing a valid visual-textual entity $E_j = (v_j, t_j)$, the system proceeds to encode these aligned entities. The visual entities are encoded into high-dimensional feature vectors using the Swin Transformer model $M_{\text{swin}}$ (Liu et al., 2021b), while both the textual entities and the event-level information are encoded using the RoBERTa model $M_{\text{RoBERTa}}$ (Liu, 2019):

$$Z_V = M_{\text{swin}}(v_j), \quad Z_T = M_{\text{RoBERTa}}(t_j) \quad Z_{event} = M_{\text{RoBERTa}}(T)$$

The encoded representations of the aligned visual entities $Z_V$, textual entities $Z_N$, and event-level information $Z_{event}$ are stored in separate Faiss indices, referred to as $Index_V$, $Index_T$, $Index_{event}$, to enable efficient retrieval.

## 3.2 Evidence Retrieval, Aggregation, and Verification

The Evidence Retrieval module is responsible for retrieving relevant entities, and event-level information from the pre-constructed Faiss index files. This module ensures efficient multimodal retrieval to support the OOC detection process. The retrieval process consists of two main components: data encoding and retrieval, followed by evidence aggregation and verification.

### 3.2.1 Evidence Retrieval

Given an input news item $N_{\text{input}} = (I_{\text{input}}, T_{\text{input}})$, where $I_{\text{input}}$ represents the image and $T_{\text{input}}$ is the accompanying caption, the system first performs entity extraction and encoding following the methods described in Sections 3.1.1 and 3.1.2. Specifically, this process results in the generation of encoded query vectors: $\mathbf{v}_{\text{query}}$ for the visual component, $\mathbf{t}_{\text{query}}$ for the textual component, and $\mathbf{e}_{\text{query}}$ for the event-level information.

Subsequently, the system retrieves the most relevant entities from the respective Meta Faiss indices by calculating the Euclidean distance between the encoded query vectors and the indexed entities. For each modality, the top two nearest entities (in terms of Euclidean distance) are retrieved. This process, referred to as **top-$k$ retrieval**, is implemented as follows:

$$\mathcal{V}_r = \text{top-}k(\mathbf{v}_{\text{query}}, Index_V, k = 2)$$
$$\mathcal{T}_r = \text{top-}k(\mathbf{t}_{\text{query}}, Index_T, k = 2)$$
$$\mathcal{E}_r = \text{top-}k(T_{\text{input}}, Index_{event}, k = 2)$$

Here, top-$k$ refers to the process of retrieving the top $k$ entities from the corresponding index $Index$, ranked by their similarity to the query vector. In this case, we set $k = 2$ to retrieve the two most relevant entities. The choice of $k = 2$ is motivated by the need to provide diverse yet concise entity representations for downstream tasks.

### 3.2.2 Evidence Aggregation and Verification

After retrieving the relevant visual entities $\mathcal{V}_r$, textual entities $\mathcal{T}_r$, and event information $\mathcal{E}_r$, these are combined by the Evidence Aggregator into a unified evidence set:

$$\mathcal{E}_{\text{agg}} = \{\mathcal{V}_r, \mathcal{T}_r, \mathcal{E}_r\}$$

The aggregated evidence $\mathcal{E}_{\text{agg}}$ is then passed to the Evidence Verifier, which assesses its consistency and relevance for the OOC detection task. The verifier ensures that there are no duplicates in the retrieved evidence and that the evidence is correctly formatted. After the verification process, $\mathcal{E}_{\text{agg}}$ is cleaned and validated, ensuring it contains only unique and properly formatted items, ready for further processing.

In summary, this process efficiently encodes and retrieves multimodal information through Faiss indices, enabling fine-grained entity-level retrieval and broader event-level context for OOC detection.

## 3.3 Multi-Agent Detection

The Multi-Agent Detection Module forms the core of our OOC detection framework, employing a multi-stage process inspired by Chain-of-Thought (CoT) reasoning. In this framework, each agent

Figure 3: Multi-Agent Detection Workflow. The system employs three agents—Retrieval, Detective, and Analyst—in a sequential pipeline, progressively refining the detection process.

is responsible for a distinct phase of the detection pipeline, with the output of one agent seamlessly feeding into the next. This enables not only a sequential but also a highly collaborative workflow, where agents complement and build upon each other's efforts. This structure closely mirrors human reasoning by breaking down complex tasks into smaller, more manageable components, allowing for a more robust and interpretable detection process.

Figure 3 outlines the roles of the three key agents in our framework: the Retrieval Agent, Detective Agent, and Analyst Agent. These agents operate sequentially to refine the detection process. The Retrieval Agent initiates fact-checking by cross-referencing input news with verified evidence, flagging any inconsistencies. The Detective Agent then conducts a deeper investigation, verifying key elements such as time, place, and objects to detect contradictions. Finally, the Analyst Agent synthesizes the previous stages' findings, providing a coherent and explainable conclusion. Through this multi-agent collaboration, MACAW not only achieves high accuracy in detecting out-of-context misinformation but also ensures that the reasoning behind each decision is transparent and interpretable. This layered, cooperative approach significantly enhances the robustness and reliability of the overall system.

### 3.3.1 RETRIEVAL AGENT

The Retrieval Agent initiates the CoT-inspired process by cross-referencing input news $N_{\text{input}}$ with retrieved evidence $\mathcal{E}_a$. It performs the first consistency check, ensuring alignment between visual and textual entities at both the entity and event levels. Leveraging MLLM's pre-trained knowledge, the agent identifies significant misalignments, passing flagged inconsistencies as input to the next agent for deeper analysis.

### 3.3.2 DETECTIVE AGENT

Building on the Retrieval Agent's flags, the Detective Agent conducts a more detailed investigation. It systematically evaluates key elements—*time*, *place*, *person*, *event*, and *object*—to detect contradictions between the retrieved evidence and the input news. For example, it checks if clothing matches the season described or if objects align with the event. This agent's refined analysis, aligned with CoT reasoning, narrows the scope of potential inconsistencies. The resulting findings are passed to the final agent.

### 3.3.3 ANALYST AGENT AND SYSTEM OUTPUT

The Analyst Agent synthesizes the outputs from the Retrieval and Detective Agents, integrating their findings into a coherent OOC detection report. Acting as an expert reviewer, it provides a well-supported, explainable conclusion, drawing on the cumulative reasoning of prior stages.

The final output of the Analyst Agent is represented as:

$$O_{\text{final}} = (C_{\text{OOC}}, T_{\text{exp}}),$$

where $C_{\text{OOC}} \in \{0, 1\}$ indicates the binary classification, with $C_{\text{OOC}} = 1$ signifying that the news is OOC, and $C_{\text{OOC}} = 0$ denoting that the news is consistent with the retrieved evidence. $T_{\text{exp}}$ provides a comprehensive explanation based on the inconsistencies and contradictions identified during the detection process. This module can facilitate structured, multi-turn dialogue by passing

outputs between agents, breaking down OOC detection tasks into manageable steps for robust and interpretable outcomes.

## 4 EXPERIMENTS

### 4.1 EXPERIMENTAL SETUP

#### 4.1.1 DATASETS

We leverage the NewsCLIPpings benchmark (Luo et al., 2021), the largest dataset for detecting out-of-context misinformation. This dataset is sourced from the VisualNews dataset (Liu et al., 2021a), which was initially created for news image captioning. NewsCLIPpings contains news articles from four major outlets: The Guardian, BBC, Washington Post, and USA Today. The dataset is evenly balanced with respect to labels.

Following prior work (Qi et al., 2024), we report results on the Merged/Balance subset, which ensures an equal distribution of retrieval strategies and positive/negative samples. Specifically, the retrieval strategies are categorized into four types: *Text-Image*, *Text-Text*, *Person Matching*, and *Scene Matching*. This subset includes 71,072 samples for training, 7,024 for validation, and 7,264 for testing. Consistent with (Luo et al., 2021), we evaluate performance using accuracy across all samples (All) and separately for the Falsified (Out-of-Context) and Pristine (Not Out-of-Context) samples as evaluation metrics.

#### 4.1.2 IMPLEMENTATION DETAILS

MACAW relies on a proprietary multi-granularity database, constructed specifically from the training subset of the NewsCLIPpings dataset. This database is built offline and comprises **18,305** unique entities and **71,072** event instances, ensuring comprehensive coverage of the training data. By pre-computing and indexing this data, we enable more efficient retrieval during inference.

To optimize retrieval efficiency, we employ a Faiss index, enabling rapid and scalable access to the multi-granularity data during the reasoning process. Each agent in the multi-agent system is instantiated using a GPT-4o-Latest architecture, with prompt engineering tailored to the specific task of each agent. This allows us to dynamically generate specialized outputs for entity recognition, event verification, and cross-modal consistency checking.

#### 4.1.3 BASELINES

To thoroughly evaluate MACAW's performance, we compare it to a broad range of SOTA multimodal models. **EANN** (Wang et al., 2018) uses adversarial training to learn event-invariant features, making it robust across various detection scenarios. **VisualBERT** (Li et al., 2019) processes image-text pairs through a unified transformer, optimizing key tasks such as image-text alignment. **SAFE** (Zhou et al., 2020) enhances prediction accuracy by transforming images into descriptive sentences and applying sentence similarity as an auxiliary loss. **CLIP** (Radford et al., 2021) employs separate encoders for images and text, aligned through contrastive learning to ensure semantically related pairs are closely represented. **CCN** (Abdelnabi et al., 2022) builds on CLIP by incorporating cross-modal consistency checks and external evidence retrieval for improved decision-making. **DT-Transformer** (Papadopoulos et al., 2023) further extends CLIP by introducing additional transformer layers to refine multimodal interactions, capturing more complex relationships. **Neu-Sym Detector** (Zhang et al., 2023a) combines neural-symbolic reasoning by decomposing text into fact queries and aggregating outputs through a pre-trained multimodal model. To demonstrate that MACAW's performance is not solely attributed to the underlying GPT-4o capabilities, we include **GPT-4o-Latest** in both zero-shot and few-shot settings as strong baselines. These variants represent the direct application of GPT-4o's multimodal capabilities without the specialized framework components present in MACAW. Finally, **SNIFFER** (Qi et al., 2024) selects the InstructBLIP as the base MLLM and enhances OOC detection with a two-stage instruction tuning process based on , integrating GPT-4-generated OOC-specific data and external evidence retrieval to improve consistency checks and overall explainability.

Table 1: Accuracy comparison (%). The best results for each column are highlighted in bold.

| Method | All | Falsified | Pristine |
|--------|-----|-----------|----------|
| EANN | 58.1 | 61.8 | 56.2 |
| VisualBERT | 58.6 | 38.9 | 78.4 |
| SAFE | 52.8 | 54.8 | 52.0 |
| CLIP | 66.0 | 64.3 | 67.7 |
| CCN | 84.7 | 84.8 | 84.5 |
| DT-Transformer | 77.1 | 78.6 | 75.6 |
| Neu-Sym detector | 68.2 | - | - |
| GPT-4o (zero-shot) | 73.8 | 75.5 | 73.4 |
| GPT-4o (few-shot) | 79.2 | 81.1 | 77.4 |
| SNIFFER | 88.4 | 86.9 | 91.8 |
| **MACAW(ours)** | **92.7** | **93.3** | **92.1** |

Table 2: Ablation Studies on Each Component of MACAW Framework.

| Analyst Agent | Detective Agent | Retrieval Agent | Multi-Grained Evidence | All | Falsified | Pristine |
|:-:|:-:|:-:|:-:|:-:|:-:|:-:|
| ✓ | ✗ | ✗ | ✗ | 83.6 | 86.3 | 80.9 |
| ✓ | ✗ | ✗ | ✓ | 82.7 | 93.1 | 72.3 |
| ✓ | ✓ | ✗ | ✓ | 89.2 | 87.5 | 90.9 |
| ✓ | ✗ | ✓ | ✓ | 88.6 | 91.0 | 86.2 |
| ✓ | ✓ | ✓ | ✓ | **92.7** | **93.3** | **92.1** |

## 4.2 MAIN RESULTS

Experimental results demonstrate MACAW's superior performance across all evaluation metrics compared to existing approaches. While traditional models trained from scratch (EANN: 58.1%, SAFE: 52.8%) and established multimodal frameworks (CLIP: 66.0%, VisualBERT: 58.6%) show limited effectiveness, more recent architectures achieve notable improvements through enhanced mechanisms. CCN (84.7%) and DT-Transformer (77.1%) leverage CLIP's foundation with additional consistency checks, while SNIFFER establishes a strong benchmark (88.4%) through its specialized detection approach. Notably, despite GPT-4o's powerful foundation and advanced reasoning capabilities, its performance peaks at 79.2% with few-shot learning—a significant improvement over its zero-shot variant (73.8%) but still substantially below MACAW's performance, highlighting the limitations of general-purpose language models for specialized detection tasks.

MACAW substantially advances the state-of-the-art with an accuracy of 92.7%, surpassing SNIFFER by 4.3% and GPT-4o (few-shot) by 13.5%. This marked improvement persists across both falsified (93.3%) and pristine (92.1%) categories, validating the effectiveness of our multi-agent reasoning framework and multi-granularity database architecture. The significant performance gap between MACAW and these strong baselines, particularly the substantial margin over GPT-4o, underscores the necessity and effectiveness of our specialized architectural design in addressing the unique challenges of OOC detection.

## 4.3 ABLATION STUDIES

To evaluate the contributions of each component in the MACAW framework, we conducted a series of ablation experiments. For setups lacking the Retrieval Agent, evidence was directly provided to the corresponding agents (e.g., Analyst Agent or Detective Agent), bypassing the retrieval process but maintaining access to multi-granularity information. This experimental design allowed us to isolate the impact of each module and demonstrate the necessity of their integration.

Starting with the Analyst Agent, which performs high-level reasoning over multi-modal inputs, MACAW achieved an initial accuracy of 83.6%. While effective in detecting falsified information, this configuration struggled with pristine content, demonstrating a clear limitation (80.9% for pristine samples). Adding the Detective Agent, responsible for fine-grained entity and image analysis, improved falsified content recall to 93.1%, but pristine accuracy dropped to 72.3%, suggesting that

Table 3: Accuracy comparison (%) between the GPT-4o and LLava Models.

|  | GPT-4o-Latest | GPT-4o-mini | LLaVA-13B | LLaVA-7B |
|---|---|---|---|---|
| **Accuracy** | 91.7% | 84.6% | 56.2% | 43.8% |

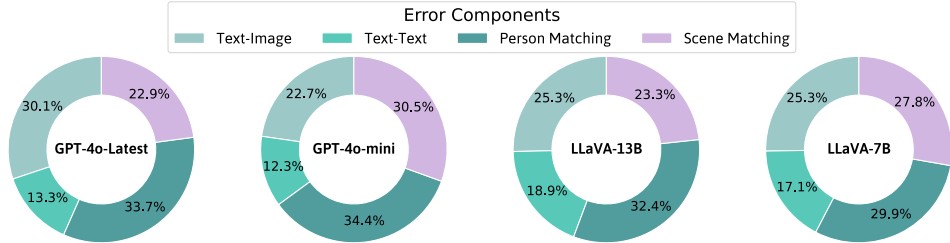

Figure 4: Error Distribution of GPT-4o and LLaVA Models on Different Type OOC Misinformation.

precise entity-level analysis alone is insufficient for balanced OOC detection. The integration of the Retrieval Agent substantially enhanced MACAW's overall performance. With this agent, the system reached an accuracy of 89.2%, with pristine content detection improving significantly to 90.9%. This highlights the critical role of external evidence in ensuring robust, contextually grounded decisions.

Finally, combining all components, including the Evidence Retrieval module, which consolidates both entity and event-level information, resulted in the highest performance. The complete MACAW system achieved an accuracy of 92.7%, with a falsified sample recall of 93.3% and a pristine accuracy of 92.1%. These results confirm that only the complete integration of all agents and multi-granularity evidence retrieval ensures optimal OOC detection performance, demonstrating the necessity of each component in achieving both high precision and generalizability.

## 4.4 DISCUSSION

### 4.4.1 WHY NOT OPEN-SOURCE MODEL?

In this section, we discuss the impact of replacing the base model in our multi-agent system with open-source alternatives. To better understand the implications of such a change, we conducted a detailed analysis using the four data types provided by the NewsCLIPpings dataset. The NewsCLIPpings dataset defines three primary types of image-text mismatches. Semantics Matching involves pairing images with captions that align in general content but differ in specific entities or events. This is split into two subtypes: ***Text-Image***, which retrieves images based on overall visual-textual similarity, and ***Text-Text***, where a semantically similar caption is first found, and the image from that caption is then mismatched with the original text. ***Person Matching*** focuses on cases where the correct individual is depicted, but the person is placed in a misleading or unrelated context. Finally, ***Scene Matching*** mislabels the broader setting or event, ensuring the environment looks similar but describes a different situation, excluding any references to individuals. For our evaluation, we maintained an equal distribution of 1,000 samples, with 250 examples from each category, to ensure a balanced and comprehensive assessment of model performance across these different misinformation scenarios.

Table 3 shows a clear performance gap between open-source models like LLaVA 1.5 (Liu et al., 2023a) and closed-source counterparts. Despite using the CLIP-ViT-L-336px architecture, LLaVA-7B and LLaVA-13B struggled with *Person Matching* and *Scene Matching* tasks, tasks requiring precise visual-textual alignment. Their smaller parameter sizes (7B and 13B) and shorter context windows limited their ability to process complex scenes. Prompt engineering yielded minimal improvements, emphasizing the architectural constraints in handling advanced multimodal reasoning.

In contrast, closed-source GPT-4o models excelled across all OOC misinformation categories, as shown in Figure 4. Their larger parameter sizes and extended context windows allowed for better handling of intricate cross-modal relationships, especially in *Scene Matching*, which requires deep contextual understanding. Additionally, the ease of deployment and regular updates of commercial models offer further advantages. Using state-of-the-art closed-source models improves the robust-

Table 4: Average rankings of four methods for logic and explanation (human and GPT-4o evaluations). The best results for each test data are highlighted in bold.

| Method | Human | | GPT-4o | |
|---|---|---|---|---|
| | Logic | Explanation | Logic | Explanation |
| LLaVA (few-shot) | 3.40 | 3.55 | 3.50 | 3.55 |
| GPT-4o (few-shot) | 2.45 | 2.33 | 2.15 | 1.73 |
| MACAW (LLaVA) | 2.90 | 3.03 | 3.08 | 3.40 |
| MACAW (GPT-4o) | **1.25** | **1.10** | **1.28** | **1.33** |

ness of our misinformation detection system while avoiding the complexities of local deployment. Continuous updates ensure that our MACAW framework remains at the forefront of multimodal misinformation detection.

### 4.4.2 EXPLAINABILITY ANALYSIS

To assess the quality of explanations generated by the MACAW framework, we conducted evaluations using both human evaluation and GPT-4o evaluation. Our comparison encompassed four configurations: two models (LLaVA-1.5-13B and GPT-4o-Latest) implemented within our MACAW framework, and the same two models applied directly in a few-shot setting. For each of the 40 randomly selected test samples, both human evaluators and GPT-4o ranked the explanations generated by the four base models according to two criteria: **logical consistency (Logic)** and **explanatory quality (Explanation)**. Each model was assigned a rank from 1 (best) to 4 (worst) for each test case, and the average ranking across all samples was calculated for both logic and explanation.

As shown in Table 6, MACAW (GPT-4o) demonstrated superior performance across all evaluation metrics, achieving the lowest average rankings in both human and GPT-4o assessments. Specifically, human evaluators assigned MACAW (GPT-4o) average rankings of 1.25 for logic and 1.10 for explanation, while GPT-4o evaluation yielded similar results with rankings of 1.28 and 1.33, respectively. These consistent results across different evaluation methods underscore the framework's robust capability in generating logically sound and clear explanations.

Interestingly, we observed that the few-shot GPT-4o baseline performed notably better than the few-shot LLaVA-1.5, suggesting the inherent strength of GPT-4o in handling OOC news detection. However, the MACAW framework substantially enhanced GPT-4o's performance, as evidenced by the improvement from few-shot GPT-4o (2.45, 2.33 for human evaluation) to MACAW (GPT-4o) (1.25, 1.10). This improvement validates the effectiveness of our framework in refining both logical reasoning and explanation generation capabilities.

A notable observation emerged when comparing human and GPT-4o evaluations of the few-shot GPT-4o method. The GPT-4o evaluator assigned more favorable scores (2.15, 1.73) compared to human evaluators (2.45, 2.33), suggesting a potential bias in GPT-4o's self-assessment. This finding aligns with previous research Wang et al. (2023); Zheng et al. (2023) on large language models' tendency toward self-favorability in evaluation tasks.

## 5 CONCLUSION

In this paper, we presented MACAW, a novel framework that combines multi-granularity retrieval with a multi-agent reasoning system to address out-of-context misinformation. Through our self-constructed database and specialized agent collaboration, MACAW demonstrates superior performance, achieving a 4.3% accuracy improvement on the NewsCLIPpings benchmark. The framework's ability to analyze multimodal inconsistencies at both entity and event levels provides a more nuanced approach to misinformation detection than existing methods. Future work could enhance MACAW by integrating external knowledge bases and extending its application to broader misinformation detection tasks, leveraging its modular architecture to develop a comprehensive solution for multimodal misinformation detection.

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

# A APPENDIX

## A.1 ERROR ANALYSIS ACROSS DIFFERENT MISINFORMATION CATEGORIES

To provide a more comprehensive understanding of MACAW's performance characteristics, we conducted a detailed analysis of error cases across different categories in the NewsCLIPpings dataset. Table 5 presents the distribution of errors across the four primary categories: ***Text-Image***, ***Text-Text***, ***Scene-Matching***, and ***Person-Matching***.

Table 5: Distribution of Error Cases Across Different Categories in NewsCLIPpings Test Dataset.

| Category | Error Count | Error Rate (%) | Primary Error Patterns |
| --- | --- | --- | --- |
| Text-Image | 177 | 33.40% | Semantic similarity confusion |
| Person-Matching | 174 | 32.83% | Contextual misalignment |
| Scene-Matching | 106 | 20.00% | Environmental ambiguity |
| Text-Text | 73 | 13.77% | Narrative consistency issues |
| Total | 530 | 100.00% | - |

Our analysis reveals several noteworthy patterns in MACAW's error distribution. Text-Image mismatches constitute the largest proportion of errors (33.40%), suggesting that the framework faces

the greatest challenges in cases where semantic similarities between images and text are subtly mis-aligned. This is closely followed by Person-Matching errors (32.83%), indicating that distinguishing individuals in different contexts remains a significant challenge despite our multi-agent approach.

Scene-Matching errors account for 20.00% of the total errors, primarily occurring in cases where environmental elements share visual similarities but represent different events or contexts. The lowest error rate was observed in Text-Text matching (13.77%), suggesting that MACAW performs relatively well in detecting inconsistencies when dealing with purely textual semantic relationships.

These findings suggest potential areas for future improvement:

1. **Enhanced semantic reasoning capabilities:** Improving the system's ability to detect subtle semantic misalignments between images and text, particularly in cases where surface-level similarities mask contextual inconsistencies.

2. **Refined person-context association:** Strengthening the framework's capability to accurately track and verify person-specific contextual information across different temporal and spatial settings.

3. **Advanced scene understanding:** Developing more sophisticated mechanisms for distinguishing between visually similar but contextually different environments and events.

4. **Improved narrative consistency checking:** Enhancing the system's ability to verify and validate textual narrative consistency across different sources and contexts.

This error distribution analysis provides valuable insights for future iterations of the MACAW framework and highlights specific areas where additional attention could yield significant improvements in overall system performance.

## A.2 FEW-SHOT PROMPTS

In our experimental setup, we carefully designed specific prompt for few-shot settings to evaluate GPT-4o model performance. The prompt was crafted to maintain consistency across different models while enabling fair comparison of their inherent capabilities. The few-shot prompts followed the examples proposed by Qi et al. (2024), incorporating two carefully selected cases that covered different types of misinformation patterns. Below, we present the exact prompts used in our experiments, maintaining their original format and structure to ensure reproducibility:

---

**Few-shot Prompt**

\# Role
You are an expert in fact-checking.
\# Task
Some news captions and accompanying images are inconsistent in terms of key news elements (5W1H) because rumormongers have taken images from other news and used them as illustrations for current news to make up multimodal misinformation. In the following examples, given the original news caption (i.e. caption_ori) for the image (i.e. image_ori), the new news caption (i.e. caption_new), and a basic description of image_ori's content, the answer was to analyze why the image and the text did not match.
You need to analyze the inconsistencies between caption_ori and caption_new in key news elements, and select from them the one most likely inconsistency between image_ori and caption_new based on the description of image_ori.

\# In-context examples
\#\# First Example
Caption_ori: John Constable's Brightwell Church and Village was part of the 2013 exhibition
Caption_new: From J Charles Eichhorn's American Skat or The Game of Skat Defined
Basic description of image_ori: This image describes a rural landscape with a farmhouse, a barn, and a field. The farmhouse is situated in the middle of the field, surrounded by the barn and the open land. The painting captures the essence of a peaceful, pastoral setting, with the farmhouse serving as the central focus of the scene.

---



The answer is: They are inconsistent in artwork. The artwork in caption_new is American Skat or The Game of Skat Defined, and the artwork in image_ori is Brightwell Church and Village. Element: artwork Entity_caption: American Skat or The Game of Skat Defined Entity_image: Brightwell Church and Village

## Second Example

Caption_ori: Chris Huhne is among the ministers expected to address delegates at next week's Lib Dem conference

Caption_new: Urs Rohner CEO of Credit Suisse participates in a panel session in Bern Switzerland on Tuesday

Basic description of image_ori: The image depicts a man wearing a suit and tie, standing at a podium with a microphone in front of him. He appears to be giving a speech or addressing an audience. In the background, there is a black screen or backdrop.

The answer is: They are inconsistent in person. The person in caption_new is Urs Rohner, and the person in image_ori is Chris Huhne. Element: person Entity_caption: Urs Rohner Entity_image: Chris Huhne

# Output Format



## A.3 EXPLAINABILITY ANALYSIS

Table 6: Average Rankings of Four Base Models for Logic and Explanation (Human and GPT-4o Evaluations). The best results for each test data are highlighted in bold.

| Method | Human | | GPT-4o | |
|---|---|---|---|---|
| | Logic | Explanation | Logic | Explanation |
| LLaVA-7b | 3.60 | 3.01 | 3.05 | 2.85 |
| LLaVA-13b | 3.20 | 3.45 | 3.55 | 3.50 |
| GPT-4o-mini | 1.90 | 2.00 | 2.00 | 2.12 |
| GPT-4o-Latest | **1.28** | **1.45** | **1.38** | **1.48** |

To assess the quality of explanations generated by the MACAW framework, we conducted evaluations using both human evaluation and GPT-4o evaluation. For each of the 40 randomly selected test samples, both human evaluators and GPT-4o ranked the explanations generated by the four base models according to two criteria: **logical consistency (Logic)** and **explanatory quality (Explanation)**. Each model was assigned a rank from 1 (best) to 4 (worst) for each test case, and the average ranking across all samples was calculated for both logic and explanation.

As shown in Table 6, GPT-4o-Latest consistently achieved the best performance, with the lowest average rankings of 1.38 for logic and 1.48 for explanation in the GPT-4o evaluation. Human evaluators provided similar results, with average rankings of 1.28 for logic and 1.45 for explanation, further confirming the model's strong reasoning capabilities and clarity. GPT-4o-mini, while slightly behind, still performed well, demonstrating the robustness of the GPT-4o architecture even in smaller-scale versions. In contrast, LLaVA-13B and LLaVA-7B performed significantly worse, with higher average rankings across both criteria. LLaVA-13B had average rankings of 3.55 for logic and 3.50 for explanation in the GPT-4o evaluation, indicating difficulties in generating coherent reasoning. LLaVA-7B also struggled, with average rankings of 3.05 for logic and 2.85 for explanation.

These results highlight the superiority of GPT-4o models in producing explanations that are both logically sound and interpretable, making them more suitable for complex multimodal reasoning tasks, such as misinformation detection.

## A.4 CASE STUDIES

Table 7 demonstrates MACAW's verification capabilities through three representative examples. Each case highlights a distinct aspect of verification: character identification in sports events, geographic consistency checking, and temporal alignment verification. These examples illustrate how MACAW conducts comprehensive analysis by leveraging multiple information dimensions beyond simple visual-textual matching.

Table 7: Case studies comparing GPT-4o and MACAW's verification capabilities across character, geographic, and temporal dimensions. Ground truth labels (GT) are provided.

**Caption**:Cleveland Cavaliers forward LeBron James dunks the ball during against the Atlanta Hawks in Game 4 of the Eastern Conference Finals. [GT: Pristine]

| **GPT-4o**: Yes, the caption matches the image. It shows a player wearing a Cleveland Cavaliers jersey with the number 23, who is dunking the basketball, and there appears to be an Atlanta Hawks player in the background. The description aligns with the visual context of the image. | **MACAW**: In give picturem, a basketball player wearing a Cleveland Cavaliers jersey number 23 and another player from the opposing team (Atlanta Hawks jersey number 4) are visible. Based on the retrieved images, we can identify one of the players arriving in **Jersey #23 as LeBron James**, a former Cleveland Cavaliers forward who played against the Atlanta Hawks in the NBA's 2015 Eastern Conference Finals. After comprehensive analysis, I think this is a *real* news. | 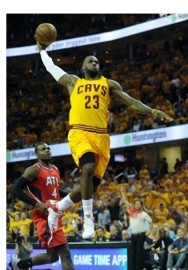 |

**Caption**: People cheer and take pictures as the pope arrives to meet with representatives of the World of Work organization in Ciudad Juarez. [GT: Falsified]

| **GPT-4o**: The caption describes people cheering and taking pictures as the pope arrives for a meeting with representatives of the World of Work organization in Ciudad Juarez. However, there is nothing in the image that definitively identifies the individual or the specific event being described. Without clear indicators of the pope's presence or explicit evidence linking the scene to Ciudad Juarez or the World of Work meeting, it is *difficult to confirm* whether the caption accurately matches the image. | **MACAW**: The image shows a crowd of people holding up phones, seemingly taking pictures or videos. A group of individuals is walking down a hallway, but no specific identifying features are visible. Based on the retrieved news caption, "The pope smiles as members of the World of Work organization wave to him in Ciudad Juarez," we can confirm the **geographic context** of the described event. However, the **individuals and locations** mentioned in the caption are noticeably absent from the image itself. This discrepancy indicates that the caption does not accurately represent the image, making it misleading or *falsified news*. | 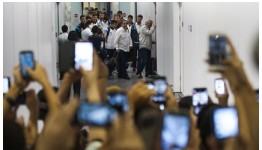 |

**Caption**: Florida Sen Marco Rubio speaks during a campaign rally in Minneapolis on Feb 23 2016. [GT: Pristine]

| **GPT-4o**: Yes, the caption appears to match the image. It shows a man holding a microphone and speaking in front of an audience, which aligns with the description of Florida Sen. Marco Rubio speaking during a campaign rally. However, to confirm if the specific date and location (Feb 23, 2016, Minneapolis) are accurate, additional context would be required. | **MACAW**: In given image, a man is speaking with a microphone in front of audiences. After comparing the retrieved image, we can confidently identify the individual as Marco Rubio. Following an event-level search, it has been confirmed that Marco Rubio participated in several presentations **during 2016**. Based on this comprehensive analysis, we conclude that this news is *real*. | 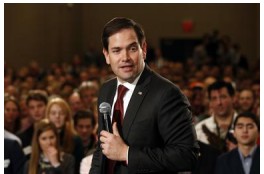 |

## A.5  ABLATION STUDY WITH GPT-4O-MINI

We conducted ablation studies with GPT-4o-mini. The experiments were performed on a balanced sample of 1,000 instances randomly drawn from the NewsCLIPpings test dataset, ensuring an equal

Table 8: Ablation Studies on Each Component of MACAW Framework with GPT-4o-mini.

| Analyst Agent | Detective Agent | Retrieval Agent | Multi-Grained Evidence | All | Falsified | Pristine |
|:---:|:---:|:---:|:---:|:---:|:---:|:---:|
| ✓ | ✗ | ✗ | ✗ | 71.3 | 67.3 | 75.3 |
| ✓ | ✗ | ✗ | ✓ | 73.5 | 70.6 | 76.4 |
| ✓ | ✓ | ✗ | ✓ | 76.3 | 75.4 | 77.3 |
| ✓ | ✗ | ✓ | ✓ | 80.6 | 79.9 | 81.4 |
| ✓ | ✓ | ✓ | ✓ | **84.6** | **85.3** | **84.0** |

distribution of falsified and pristine samples. The results, presented in Table 8, provide insights into how each component of the MACAW framework contributes to overall performance when GPT-4o-mini is employed.

As shown in Table 8, the overall performance of GPT-4o-mini is noticeably lower compared to the full GPT-4o model. The highest accuracy achieved with GPT-4o-mini is 84.6%, which is significantly below the 92.7% obtained with GPT-4o (see Table 2). This performance gap can be largely attributed to GPT-4o-mini's limitations, including its reduced context window and weaker reasoning capabilities. These factors restrict its ability to handle complex multimodal tasks, particularly in cases where detailed cross-modal reasoning is required. This limitation is particularly evident in the "Falsified" category, where reasoning-intensive detection tasks are more prevalent, leading to a lower recall rate for falsified news.

Despite the limitations of GPT-4o-mini, the introduction of multiple agents led to consistent improvements in performance. When the Analyst Agent was used alone, the system achieved an accuracy of 71.3%. Adding the Detective Agent raised the accuracy to 76.3%, highlighting the importance of multi-agent collaboration in decomposing the verification task. The combination of both agents allowed the system to better capture inconsistencies across different modalities, although the gains were more modest compared to the full GPT-4o model due to the base model's inherent limitations.

Among the various agents, the Retrieval Agent had the most significant impact on performance. When this agent was introduced, the overall accuracy increased to 80.6%, marking the largest single-agent improvement. The Retrieval Agent's role in performing a preliminary analysis of the image content proved crucial, as it helped to reduce the error rate by retrieving relevant visual and textual evidence early in the detection pipeline. This analysis mitigated some of the weaknesses of GPT-4o-mini, particularly in tasks requiring the initial disambiguation of multimodal content.

In conclusion, these ablation studies demonstrate that while GPT-4o-mini serves as a functional alternative, it cannot match the performance of the full GPT-4o model in handling the complex reasoning and cross-modal verification tasks required by the MACAW framework. Nevertheless, the multi-agent architecture and retrieval mechanisms of the framework continue to provide substantial improvements, even when a smaller base model is used. This highlights the robustness of the MACAW framework, which remains effective across different model scales.

## A.6    COMPARISON WITH SNIFFER MODEL

In this section, we provide a detailed comparison between our proposed MACAW framework and the SNIFFER model Qi et al. (2024), a prominent approach in the field of OOC misinformation detection. Both models leverage the power of MLLMs to tackle the challenges of OOC misinformation, yet they differ significantly in methodology, performance, explainability, and adaptability to various datasets, leading to distinct advantages and limitations.

From a methodological perspective, SNIFFER employs a two-stage instruction tuning approach, adapted from InstructBLIP, to refine its ability to align generic objects with news-domain entities and subsequently fine-tune its discriminatory powers for OOC misinformation detection. This process involves the integration of external knowledge through retrieval mechanisms, enabling SNIFFER to perform both internal checks (image-text consistency) and external checks (claim-evidence relevance), with the final decision produced through composed reasoning. While this is an effective approach, it introduces a reliance on external retrieval systems, which can introduce noise and latency in real-time applications. In contrast, MACAW adopts a multi-agent architecture that decomposes the complex reasoning task into specialized subtasks, handled by agents responsible for retrieval, detection, and

analysis. This modular structure not only enhances the interpretability of the system but also allows for more fine-grained verification through multi-granularity retrieval of both entity- and event-level information. By structuring its framework around a self-constructed multi-granularity database, MACAW reduces dependency on external sources, offering a more efficient and unified approach to misinformation detection.

In terms of performance, both models demonstrate state-of-the-art capabilities, but MACAW consistently outperforms SNIFFER across several benchmarks. SNIFFER reports an accuracy of 88.4% on the NewsCLIPpings dataset, leveraging its external retrieval mechanisms to detect inconsistencies in OOC samples. However, MACAW achieves an accuracy of 92.7%, a significant improvement attributed to its multi-agent collaboration and multi-granularity retrieval system. This structured approach allows MACAW to handle more subtle and complex OOC cases by cross-validating information across different granularities, thus providing a more robust detection mechanism. While SNIFFER's retrieval-based methodology strengthens its performance, particularly in cases where external evidence is readily available, MACAW's internal verification process ensures that it remains highly effective even in scenarios where such evidence may be limited or noisy.

Explainability is another critical dimension where the two models diverge. SNIFFER integrates its internal and external verification results to generate explanations, often relying on external evidence to justify its decisions. By incorporating web-based evidence, SNIFFER can provide detailed explanations that highlight the inconsistencies between the image and the text, such as misidentified entities or mismatched events. However, this reliance on external data can sometimes lead to overfitting to retrieved evidence, potentially complicating the interpretability of the decision-making process. MACAW, on the other hand, enhances explainability through its multi-agent architecture, where each agent contributes specialized reasoning to the final output. The Retrieval Agent, Detective Agent, and Analyst Agent collaborate to ensure that the reasoning process is transparent and interpretable at every stage. By ensuring that the decision-making process is broken down into distinct phases, MACAW not only provides accurate judgments but also offers more structured and comprehensible explanations, further strengthened by the integration of multi-granularity data, which adds depth to its contextual understanding.

When considering the adaptability of these models to diverse datasets, MACAW's design offers a clear advantage. SNIFFER demonstrates strong generalization capabilities, as evidenced by its success across datasets such as News400 and TamperedNews, where it outperforms several baselines. However, its reliance on external retrieval introduces potential vulnerabilities to noisy or incomplete data, which can affect its overall robustness. MACAW's multi-granularity database construction and internal verification process allow it to adapt more effectively to different types of misinformation across various contexts. By cross-referencing data at both the entity and event levels, MACAW ensures that it can consistently maintain high performance across diverse datasets without being overly dependent on the availability of external evidence. This adaptability makes MACAW particularly well-suited for real-world applications where external sources may not always provide reliable or timely information.

Finally, with respect to efficiency, MACAW's multi-agent system provides a significant advantage. SNIFFER's reliance on external tools and web-based retrieval can introduce latency, particularly in real-time or large-scale applications where the availability and quality of external data are critical. In contrast, MACAW's internal multi-agent collaboration and self-constructed database allow it to operate more efficiently. The modular design of MACAW's agents ensures that each step of the verification process is optimized for speed and accuracy, making it more suitable for real-time OOC misinformation detection. By reducing dependency on external retrieval, MACAW minimizes computational overhead while maintaining high detection accuracy, a crucial factor for practical deployment in fast-paced information environments.

In conclusion, while both SNIFFER and MACAW represent significant advancements in the detection of OOC misinformation, MACAW's innovative multi-agent architecture, multi-granularity retrieval system, and focus on internal verification offer superior performance, interpretability, and adaptability. These differences highlight MACAW's robustness in handling complex misinformation scenarios and its potential for real-world application, setting it apart as a more comprehensive and efficient solution for OOC misinformation detection.

