# OpenReview forum: "Detecting Out-of-Context Misinformation via Multi-Agent and Multi-Grained Retrieval"
_ICLR.cc/2025/Conference — ICLR 2025 Conference Withdrawn Submission_

### Official Review · Reviewer_pS43 · 2024-10-17

**Soundness:** 2
**Presentation:** 3
**Contribution:** 2
**Rating:** 3
**Confidence:** 4

**Summary:**

This paper introduces MACAW, a framework designed to identify out-of-context (OOC) misinformation through three key stages: "EVIDENCE STORAGE," "EVIDENCE RETRIEVAL, AGGREGATION, AND VERIFICATION," and "MULTI-AGENT DETECTION."

1. Initially, MACAW establishes an "evidence storage" containing multi-model evidence at the entity level and textual evidence at the event (caption) level.

2. Then, with information that needs to be detected, MACAW retrieves relevant evidence based on similarities.

3. Finally, MACAW employs a multi-agent (multi-step) design approach, prompting GPT-4o to focus on specific steps (Relevance, Temporal, Spatial, Object, and Event) to provide a final judgment gradually.

Experimental results demonstrate that MACAW outperforms other approaches on the NewsCLIPpings dataset.

**Strengths:**

1. This paper is well-structured and easy to read.
2. The "evidence storage" in MACAW presents a relatively novel approach. The authors propose multimodel entities that comprise textual entities and image entities with efforts to ensure alignment between the two modalities.

**Weaknesses:**

1. Vague Motivation: The motivation presented in this paper lacks clarity. The authors claim: "We developed the first multi-agent OOC detection system that integrates multi-granularity information, mirroring the real-world collaboration among human experts." However, as far as I know, human experts responsible for misinformation detection typically make independent judgments and reach a final consensus through discussion, rather than each expert being responsible for only a single part of the detection process. In contrast, MACAW assigns each agent responsibility for only one specific step of the detection process. This distinction seems misaligned with how human experts collaborate.
   https://www.politifact.com/article/2018/feb/12/principles-truth-o-meter-politifacts-methodology-i/

2. Lack of Literature Review: The paper overlooks some relevant works that focus on enhancing explainability, which should be acknowledged. This includes, but is not limited to:
   - Interpretable Multimodal Misinformation Detection with Logic Reasoning (Findings of ACL 2023)
   - Interpretable Detection of Out-of-Context Misinformation with Neural-Symbolic-Enhanced Large Multimodal Model (Findings of ACL 2023)
   - Interpretable Multimodal Out-of-context Detection with Soft Logic Regularization (ICASSP 2024)

3. Insufficient Details on Data Storage: The paper lacks sufficient detail on how the data storage is constructed. What are the data sources? Does it retrieve information from the internet? If so, have the authors accounted for potential time leakage during retrieval? For instance, some misinformation may have already been flagged as fake news by the time it's retrieved from the web. This could introduce unintended biases, particularly when handling older misinformation cases.

4. Incomplete Experimental Reports: The experimental results are incomplete. Table 1 only shows results on the Merged/Balance subset of the NewsCLIPpings dataset. What about other subsets, such as Text-Image, Text-Text, Person-Matching, and Scene-Matching? A broader range of experiments would give a clearer picture of the strengths and weaknesses of MACAW framework.

5. Unreasonable Explainability Analysis: Section 4.4.2, Explainability Analysis, is insufficient. The paper compares different backbone models within MACAW framework, but it should instead provide a comparison with other OOC detection approaches in terms of explainability. The current comparison only shows how GPT-4o performs better than other models, which is not enough to assess the explainability of the system. Additionally, incorporating more qualitative analysis in this section would further strengthen the evaluation.

**Questions:**

1. The evidence source description is vague. How did the authors address the issue of time leakage? Without clear details on data retrieval, how can we be sure the model isn’t accessing information that wasn’t available at the time of the misinformation event?

2. The comparison models are mainly based on CLIP, which has far fewer parameters and less training data than GPT-4o. Are the performance gains from MACAW due to the method itself, or simply the use of a much larger model like GPT-4o? Moreover, given GPT-4o’s massive training data, is there a risk that NewsCLIPpings data was included, leading to data leakage?

I am open to revising my score once the authors address my concerns.

---

### Official Review · Reviewer_QvfK · 2024-11-01

**Soundness:** 3
**Presentation:** 3
**Contribution:** 3
**Rating:** 5
**Confidence:** 4

**Summary:**

The paper addresses the challenge of misinformation caused by out-of-context image-text pairs. The authors propose and implement MACAW (Multi-Agent Cross-Modal Misinformation Analysis Workflow), which leverages large vision-language models (such as GPT-4o-latest, GPT-4o-mini, LLaVa-7B and LLaVa-13B), multi-agent and multi-grained retrieval. Specifically MACAW comprises:
1) A Retrieval Agent tasked with gathering relevant “evidence” (visual entities, textual entities, and events) from a proprietary, pre-constructed database. This agent conducts an initial analysis to identify inconsistencies between the retrieved evidence and its relationship to the image-text pair under review.
2) A Detective Agent performs a deeper analysis of contextual elements (time, location, individuals, events, and objects) and works to identify any inconsistencies within these aspects.
3) An Analyst Agent responsible for analyzing the outputs from the previous agents in order to deliver a final verdict and explanation.

The authors conduct experiments on the NewsCLIPpings dataset, providing a comparative analysis against state-of-the-art (SotA) methods, where the proposed MACAW method with GPT-4o-Latest achieves superior performance. Additionally, an ablation study highlights the contribution of each component, accompanied by an evaluation of the model’s generated explanations and logic.

**Strengths:**

The paper is overall well-written and easy to follow.
It presents a novel pipeline for detecting out-of-context misinformation using multiple agents, achieving high performance on the NewsCLIPpings dataset.
The ablation study demonstrates the necessity of each agent and component.
The method focuses on interpretability which is crucial especially when developing tools intended for use by the general public or journalists.

**Weaknesses:**

W1) The authors utilize the NewsCLIPpings dataset, which includes ‘out-of-context’ samples that are ‘synthetic’ or algorithmically created by mismatching the original image-text pairs. However, strong performance on ‘synthetic’ data does not guarantee similar results on real-world data. The absence of evaluation against real-world out-of-context misinformation is a notable limitation.

W2) There is no comparison with the current SotA method on NewsCLIPpings such as the “Attentive Intermediate Transformer Representations” (AITR) model [1] which achieves 93.3% on NewsCLIPpings. Moreover, the authors of [1] express concerns regarding the use of the NewsCLIPpings dataset, noting how models can achieve high performance on it by relying on superficial patterns rather than factuality.

W3) While the ablation study demonstrates that each component of the proposed method is important, Table 3 reveals the significance of using the more powerful GPT-4o-latest (92.7%) over GPT-4o-mini, which achieves only 84.6%. This performance is surpassed by previous methods, such as CCN (84.7%) and SNIFFER (88.4%). Consequently, this somewhat diminishes the significance of the proposed method, suggesting that the observed performance improvement is primarily attributable to the more powerful LVLM rather than the method itself.

W4) Table 4 presents the ranking of each model’s explanations and logical consistency. However, this does not necessarily indicate the quality of the explanations, just that they are preferred over the explanations of other methods.

W5) The presentation of competing methods is somewhat superficial, lacking an in-depth discussion of how the proposed method distinguishes itself from other LVLM-based approaches, such as SNIFFER or [2, 3].

References
[1] Papadopoulos, S. I., Koutlis, C., Papadopoulos, S., & Petrantonakis, P. C. (2024). Similarity over Factuality: Are we making progress on multimodal out-of-context misinformation detection?. arXiv preprint arXiv:2407.13488.
[2] Tahmasebi, S., Müller-Budack, E., & Ewerth, R. (2024, October) Multimodal Misinformation Detection using Large Vision-Language Models. In Proceedings of the 33rd ACM International Conference on Information and Knowledge Management (pp. 2189-2199).
[3] Geng, J., Kementchedjhieva, Y., Nakov, P., & Gurevych, I. (2024). Multimodal Large Language Models to Support Real-World Fact-Checking. arXiv preprint arXiv:2403.03627.

**Questions:**

While I believe this paper holds valuable potential for the research community, certain limitations need to be addressed. My suggestions for the authors:

Q1) Evaluate WACAW against real-world out-of-context data, such as the VERITE benchmark (https://github.com/stevejpapad/image-text-verification) to address W1.

Q2) An evaluation on the “Miscaptioned Images” subset of VERITE would also be very valuable to demonstrate that the proposed method is effective even when the text contains inaccuracies, such as altered locations, dates, or individuals. This evaluation could partially address concerns raised in W2.

Q3) Expand the ablation study (Table 2) to include an evaluation of GPT-4o-mini, demonstrating that while the high performance appears primarily due to the capabilities of GPT-4o-latest, the proposed WACAW method consistently enhances performance across models. This could partially address W3.

Q4) The paper would benefit from the inclusion of inference examples, showcasing both correct and mistaken predictions along with their explanations. This could partially address W4.

Q5) Based on the supplementary material, it appears that the “proprietary multi-granularity database” primarily utilizes portions of the NewsCLIPpings dataset as its “event instances” and extracts visual and textual entities for the entity databases. Is this correct? If so, why isn’t this made explicit in the paper? Additional information about the database and its design is necessary, including a clarification on whether the authors have ensured there is no data leakage between the training, validation, and test sets.

Q6) Table 1 could also indicate which of these methods leverage: external information from the web and/or a knowledge database and/or visual/textual entities and/or LVLMs, etc., so as to provide a more fair and informative comparison. Alternatively, you can mention this information in section 4.1.3.

---

### Official Review · Reviewer_MBt7 · 2024-11-03

**Soundness:** 3
**Presentation:** 3
**Contribution:** 3
**Rating:** 5
**Confidence:** 4

**Summary:**

The approach in this paper is feasible for enhancing OOC detection accuracy. However, the work required to build the evidence module may lack originality, potentially limiting its impact on future work in this area.

**Strengths:**

This solution is intuitive and supported by a self-built multi-granularity evidence storage module. This module integrates information at various levels, such as entities and events, providing a basis for detecting and interpreting anomalies. When the evidence storage module is sufficiently large and up-to-date, it will significantly improve the accuracy of OOC detection.

**Weaknesses:**

1. The core of this work should focus on the accuracy of image-text matching in the evidence stored within the module and the quantity of evidence it contains. Moreover, to ensure high accuracy in OOC detection, the evidence storage module may need constant updates; otherwise, maintaining a high detection accuracy for OOC over time may be challenging. Consequently, this might limit the work’s impact on future OOC research.
2. In the example of image-text detection shown in Figure 1, the image depicts Cameron wearing fall-winter attire, with the caption stating that Cameron left the High Court on June 14. This discrepancy is quite evident, appearing solvable without extensive background data. Figure 2 revisits this example but  it still relies on empirical information for news detection.

**Questions:**

1. Are there examples where accurate judgments are made based on the caption’s time, character, or location? The examples in the paper seem solvable by individuals without background knowledge or the multi-agent support mentioned.

---

### Note · Authors · 2024-11-27

**Comment:**

Thank you to the reviewers for their valuable commentss, to the one reviewer who responded positively to me, and to the two reviewers who ignored me.

**Withdrawal Confirmation:**

I have read and agree with the venue's withdrawal policy on behalf of myself and my co-authors.